# Optimization of Porphyran Extraction from *Pyropia yezoensis* by Response Surface Methodology and Its Lipid-Lowering Effects

**DOI:** 10.3390/md19020053

**Published:** 2021-01-23

**Authors:** Dan He, Liping Yan, Yingxia Hu, Qifang Wu, Mingjiang Wu, Jong-il Choi, Haibin Tong

**Affiliations:** 1College of Life and Environmental Science, Wenzhou University, Wenzhou 325035, China; hed1990@stu.wzu.edu.cn (D.H.); yanlp@stu.wzu.edu.cn (L.Y.); huyx@stu.wzu.edu.cn (Y.H.); wuqf@stu.wzu.edu.cn (Q.W.); 2Department of Biotechnology and Bioengineering, Chonnam National University, Gwangju 61186, Korea

**Keywords:** *Pyropia yezoensis*, porphyran, response surface methodology, lipid-lowering effect

## Abstract

Macroalgae polysaccharides are phytochemicals that are beneficial to human health. In this study, response surface methodology was applied to optimize the extraction procedure of *Pyropia yezoensis* porphyran (PYP). The optimum extraction parameters were: 100 °C (temperature), 120 min (time), and 29.32 mL/g (liquid–solid ratio), and the maximum yield of PYP was 22.15 ± 0.55%. The physicochemical characteristics of PPYP, purified from PYP, were analyzed, along with its lipid-lowering effect, using HepG2 cells and *Drosophila melanogaster* larvae. PPYP was a β-type sulfated hetero-rhamno-galactan-pyranose with a molecular weight of 151.6 kDa and a rhamnose-to-galactose molar ratio of 1:5.3. The results demonstrated that PPYP significantly reduced the triglyceride content in palmitic acid (PA)-induced HepG2 cells and high-sucrose-fed *D. melanogaster* larvae by regulating the expression of lipid metabolism-related genes, reducing lipogenesis and increasing fatty acid β-oxidation. To summarize, PPYP can lower lipid levels in HepG2 cells and larval fat body (the functional homolog tissue of the human liver), suggesting that PPYP may be administered as a potential marine lipid-lowering drug.

## 1. Introduction

The ocean contains various resources, comprising functional algae, such as brown algae, red algae, green algae, and coralline algae, which are favorable candidates for the development of high value-added products [1,2]. Marine algae are abundant sources of medicinal phytonutrients that demonstrate beneficial effects on human health. Among them, brown and red algae contain polysaccharides with antioxidation, anti-inflammatory, antitumor, and immunomodulatory activities such as fucoidan, carrageenan, and alginate, which are used as functional foods [3,4,5]. Healthy foods are the basis for a healthy life and help to prevent the occurrence of noncommunicable diseases including diabetes, cardiovascular diseases, certain types of cancer, and nonalcoholic fatty liver disease (NAFLD) [6]. The official data of the World Health Organization state that an estimated 2 billion people lack access to safe, nutritious, and sufficient amounts of food worldwide. A surge in the production and consumption of highly processed food, rapid unplanned urbanization, and changing lifestyles have also contributed to increased consumption of an unhealthy diet, whose components are energy-dense (with high-fat and free sugar contents). As is known, the liver plays a major role in the regulation of fat and carbohydrate metabolism. However, a high-fat or high-sugar diet alters the normal metabolic process, commonly leading to the accumulation of triglycerides (TGs) within hepatocytes and to a clinical condition known as NAFLD [7,8,9]. NAFLD is a chronic liver disorder with increasing prevalence owing to the global epidemic of obesity not only among the middle-aged and older adults [10], but also among the youth and children [11,12].

The “two-hit” theory of NAFLD is based on the two steps of liver injury: intrahepatic lipid accumulation, and inflammatory progression to nonalcoholic steatohepatitis [13,14,15,16]. Therefore, reducing lipid accumulation is an effective means to block the progression of NAFLD at an early stage. Of note, palmitic acid (PA), a saturated fatty acid, is usually found in animal fats and vegetable oils. Owing to its abundance, it is easy to consume more than the recommended intake, which can induce lipotoxicity in hepatocytes [17,18]. In addition, high-sugar diets such as processed foods and beverages promote de novo lipogenesis in the liver, leading to TG accumulation, which in turn exacerbates NAFLD [19,20,21]. There is considerable evidence that PA-induced HepG2 cells and high-sucrose-fed *D. melanogaster* can be used to mimic the model of lipid accumulation-related disease for screening potential therapeutic lead compounds [22,23,24,25,26]. 

*Pyropia yezoensis*, rich in porphyran, is a kind of marine vegetable that is widely distributed and popular in China, Korea, and Japan. Studies have shown that porphyran derived from *P. yezoensis* demonstrates some pharmacological properties such as anti-inflammatory [27], antitumor [28], antioxidation, immunomodulation, anticardiovascular, and anticerebrovascular functions [29]; however, the lipid-lowering effect of *P. yezoensis* porphyran remains unclear. In this study, the yield of porphyran isolated from *P. yezoensis* via hot water extraction was optimized using response surface methodology (RSM) with a powerful statistical and mathematical model. In addition, the lipid-lowering effects of PPYP, purified from porphyrin (PYP), and the underlying mechanism were investigated using two distinct high calorie-induced models in vitro and in vivo. The present study examining on the extraction process and its bioactivity laid the foundation for the development and application of a potential marine drug or functional foods.

## 2. Results

### 2.1. Single-Factor Experiment Analysis

The effects of the time, temperature, and liquid–solid ratio on the yield of PYP were investigated via the single-factor experiments (Figure 1). The yield of PYP was affected by the liquid–solid ratio (10:1, 20:1, 30:1, 40:1, and 50:1 mL/g), and the yield increased as the liquid–solid ratio increased from 10:1 to 30:1 mg/L, reaching a maximum rate at a liquid–solid ratio of 30:1 (Figure 1A). As shown in Figure 1B, the PYP yield increased rapidly as temperature increased from 60 to 100 °C, and reached a maximum of 18% at 100 °C. In accordance with the present result, 90 °C was determined as the center-spot for optimum temperature. In addition, the PYP yield gradually increased from 30 to 120 min and reached a maximum of 17.7% at 120 min, without noticeable change from 120 to 240 min (Figure 1C). However, the PYP yield reached 17.1% at 60 min, almost equivalent to that observed at 120 min. Therefore, the time center point should be between 30 and 120 min. Therefore, the duration ranging from 30 to 120 min was selected as the time factor for the RSM experiment.

### 2.2. Response Surface Analysis 

Three individual extraction parameters were optimized by analyzing the response values for the 17 trials via a Box–Behnken design (BBD). As shown in Table 1, the PYP yield obtained after the trials ranged from 9.96 to 20.28%; these data were analyzed using the Design-Expert software. The second-order polynomial equation was then used to establish the relationship between the PYP yield and the three variables as follows: Y = 14.34 + 1.48A + 1.20B + 3.41C + 1.53AB − 0.065AC − 0.052BC − 1.58A^2^ − 0.88B^2^ + 2.68C^2^
where Y: PYP yield (%); A: liquid–solid ratio; B: temperature; C: time.

As shown in Table 2, the model F-value was 254.25 with a significant *p*-value (<0.0001), indicating that the model could be applied to predict the yield of PYP. R-squared (0.9970), adj R-squared (0.9930), pred R-squared (0.9811), and coefficient of variation (1.90%) also proved that the experimental model was accurate. 

As shown in Figure 2 and Appendix A, the relationship between independent and dependent variables was analyzed using the Design-Expert software. The 3-D response surface and 2-D contour plots show the interaction between the two variables, as well as the relationship between the response of each variable and the experimental level. Figure 2A and Appendix A show the joint influence of the liquid–solid ratio and time on the PYP yield at a temperature level of 0. The shapes of the contour plots shown in Appendix A are nearly elliptical, indicating that the interactions of time and temperature were significant to the PYP yield, which is consistent with the regression coefficient significance obtained from the equation. Although the response surfaces shown in Appendix A were steep, Figure 2B,C were not elliptical, suggesting that the temperature and liquid–solid ratio did not exhibit significant mutual interactions (Figure 2B and Appendix A). The results for the interaction of time and temperature showed a similar trend (Figure 2C and Appendix A). The aforementioned analysis of the response surface demonstrated that the liquid–solid ratio and time had a significant effect on the PYP yield. 

The optimal conditions for the extraction of PYP obtained using the model equation are listed as follows: temperature, 100 °C; time, 120 min; liquid–solid ratio, 29.32 mL/g; the maximum predicted yield of PYP was 22.05 ± 0.37%. Validation experiments were performed to verify the predicted yield of PYP for the optimal conditions, and the actual experimental yield of PYP was 22.15 ± 0.55%, which was not significantly different from the predicted yield. This result indicates that the analytical model of this study can be applied to predict the extraction conditions of PYP.

### 2.3. Purification of PYP and Its Physicochemical Properties

PYP was deproteinated by the Sevag method, and the final product was named PPYP. The physicochemical properties of PPYP are shown in Table 3. The total sugar content of PPYP was determined to be 93.2 ± 1.5%, composed mainly of galactose and rhamnose, with a molar ratio of 5.3:1 (Appendix A.). PPYP contained a sulfate group, indicating that PPYP was a negatively charged hetero-rhamno-galactan. HPGPC profile of PPYP showed a sharp peak (Appendix A) with an average molecular weight of 151.6 kDa, which is consistent with a previous report [30]. The results demonstrated that PPYP was a sulfated hetero-rhamno-galactan. 

### 2.4. FT-IR Spectroscopy Characteristics of PPYP 

The characteristics of PPYP were examined by Fourier transform infrared (FT-IR) spectrometer (Figure 3). The peaks at 3448, 2935, and 1637 cm^−1^ presented the stretching vibrations of hydroxyl, alkyl, and carboxyl groups [31], respectively. The stretching vibration of S=O exhibited a peak near 1240 cm^−1^ which was characterized by the presence of sulfate groups [32]. The characteristic absorption peak of the ether bond (-C-O-C-) was observed at 931 cm^−1^, indicating that PPYP contains 3,6-anhydro-α-l-galactose [33], which is consistent with the physicochemical property of PPYP. The peak at 1000–1200 cm^−1^, 500–800 cm^−1^, and 891 cm^−1^ confirmed that PPYP existed in the pyran structure with a β-type glycosidic bond. PPYP also showed a peak of Gal at 1074 cm^−1^ [34,35,36,37]. The FT-IR spectra illustrated that PYPP might be a β-type sulfated galactan-pyranose.

### 2.5. Effect of PPYP on Lipid Accumulation in Palmitic Acid (PA)-Induced HepG2 Cells 

#### 2.5.1. Cytotoxicity of PPYP and PA on HepG2 Cells 

HepG2 cells were treated in different concentrations of PPYP or PA for 48 h. When the cells were treated with PA at the concentrations ranging from 25 to 200 μM, no cytotoxicity toward HepG2 cells was observed compared with that in the normal control (NC) group (Figure 4A). As shown in Figure 4B, PPYP had no effect on cell viability in the concentration range 12.5–200 μg/mL. Therefore, the concentrations of PA up to 200 μM and PPYP at 200 μg/mL were chosen for the following experiments. 

#### 2.5.2. PPYP Alleviates PA-Induced TG Accumulation in HepG2 Cells 

PA was used to induce TG accumulation in HepG2 cells. As shown in Figure 5A, compared with the NC group, the TG content increased in a PA concentration-dependent manner. Treatment with 200 μM PA demonstrated the highest TG content in HepG2 cells. Therefore, 200 μM was selected as the appropriate concentration of PA to develop a cellular model of TG accumulation in vitro. PPYP supplementation significantly decreased the TG accumulation in HepG2 cells compared to that in the PA group (200 μM, Figure 5A). Oil red O staining also revealed that PPYP significantly decreased the hepatic intracellular TG accumulation (Figure 5B). Based on these results, the lipid metabolism-related gene was analyzed via qPCR. As shown in Figure 5C, the levels of sterol regulatory element binding transcription factor 1 (*SREBP1*), acetyl-CoA carboxylase (*ACC*), and fatty acid synthase (*FAS*) were decreased in the PPYP + PA group compared to that in the PA group. In addition, we found that the expression of carnitine palmitoyltransferase 1 (*CPT1*) and peroxisome proliferator activated receptor alpha (*PPARɑ*) increased upon treatment with PPYP, suggesting that PPYP could trigger fatty acid β-oxidation and inhibit lipogenesis.

### 2.6. Effect of PPYP on Lipid Accumulation in High-Sucrose-Fed D. melanogaster Larvae 

*D. melanogaster* larvae were cultured in normal medium containing PPYP. The data depicting body weights are shown in Figure 6A. PPYP did not cause a significant difference in the body weight, as well as TG content, in third instar larvae compared to that observed in the NC group, suggesting that PPYP did not affect physiological lipid content in the larvae fed with a normal diet. However, a high-sucrose-diet (HSD) significantly promoted TG accumulation in third instar larvae. Compared to the HSD group, the PPYP + HSD group demonstrated a significant reduction in the TG content (Figure 6B). The aforementioned results indicate that PPYP exhibited a potential lipid-lowering ability.

To determine the potential mechanisms by which PPYP alleviated the elevated TG in HSD-fed larvae, the expression of several lipid synthesis-related genes in the fat body of third instar larvae were examined using qPCR. FAS is a lipogenic factor activated by the transcription factor SREBP. Compared to the HSD group, *FAS* expression decreased (0.8-fold) in the PPYP + HSD group; *SREBP* expression also decreased significantly (0.6-fold), indicating that PPYP prevented an increase in lipogenesis after the administration of a high-calorie diet (Figure 6C). We further determined the expression of genes involved in lipid catabolism, the expression of acyl-Coenzyme A oxidase at 57D distal (*Acox57D-d*), fatty acid-binding protein (*FABP*), and phosphocholine cytidylyltransferase 1 (*CCT1*) tended to decrease in the HSD group. PPYP supplementation significantly upregulated the expression of *Acox57D-d* and *FABP* compared to that observed with HSD alone (Figure 6C). These results suggest that PPYP reduced lipid accumulation by regulating the expression of genes associated with lipid metabolism.

## 3. Discussion

The extraction process plays a crucial role in the study of porphyran, as the process parameters influence the polysaccharide yield, structure, and bioactivity [38]. Different extraction technologies, including hot water extraction, microwave-assisted extraction, ultrasonic-assisted extraction, and enzyme-assisted extraction, have been used to obtain polysaccharides from macroalgae [39,40]. Among them, hot water extraction is considered the conventional method owing to its simplicity and safety. The main influencing factors include extraction temperature, extraction time, and liquid–solid ratio. Limited information is available on the optimal conditions for the hot water extraction of PYP. Using RSM, the present study showed that the maximum yield of PYP was 22.15 ± 0.55%, obtained using hot water extraction. PPYP derived from PYP was analyzed; it was a β-type sulfated hetero-rhamnogalactan-pyranose with medium molecular weight. Accumulating evidence demonstrates that bioactive polysaccharides inhibit diet-induced metabolic disease by inhibiting lipid accumulation, including NAFLD [41] and obesity [42]. We noted that high-caloric diets easily promoted lipid accumulation, and subsequently led to the development of metabolic diseases. Hence, we investigated whether PPYP also inhibited lipid accumulation.

High-fat or high-sugar diets exert adverse effects on liver lipid metabolism, which is characterized by the deposition of TG as lipid droplets in the cytoplasm of hepatocytes [43]. HepG2 cells display many genotypic features of normal human hepatocytes [44] and are widely used to evaluate hepatic function in vitro [45]. According to a previous study, the PA-induced HepG2 cell hepatosteatosis model is commonly used that identify bio-active compounds to inhibit lipid accumulation. For example, Zhong et al. demonstrated that the *Ganoderma lucidum* polysaccharide peptide (GLPP) reduced the accumulation of lipid droplets and the content of TG in the hepatosteatosis model of HepG2 cells [46]. Huang et al. reported the preventive and therapeutic effects of resveratrol on PA-induced hepatocyte steatosis in HepG2 cells [47]. Therefore, in this study, the PA-induced HepG2 cells hepatosteatosis model was utilized for evaluating the lipid-lowering ability of PPYP. Consistent with previous studies [48], our study showed that PA could significantly increase lipid accumulation in a dose-dependent manner in HepG2 cells. PPYP significantly reduced TG content, indicating the inhibition of the PA-induced steatosis in HepG2 cells. SREBP1, ACC, and FAS play an important role in regulating the synthesis of TG [49]. PPYP could downregulate the gene expression of *SREBP1*, *ACC*, and *FAS*, consistent with the decrease in TG content. Additionally, the expression levels of *CPT1* and *PPARα*, regulators of fatty acid β-oxidation, were significantly increased upon PPYP supplementation [50]. The result demonstrated that PPYP might reduce lipid accumulation by increasing fatty acid β-oxidation. Therefore, PPYP contributes to the enhancement of fatty acid β-oxidation and inhibition of lipogenesis in hepatocytes. Lipid accumulation in hepatocytes leads to hepatic steatosis, which is an early feature of NAFLD [51]. Therefore, we speculated that PPYP might prevent the progression of NAFLD by reducing lipid accumulation. 

Recent evidence suggests that sugar also promotes fat accumulation and further increases the risk of NAFLD [52]. Sucrose is commonly used as a sweetener worldwide. Intake of sucrose-sweetened beverages (SSBs) enhances the levels of circulating TGs and enhanced de novo lipogenesis in the liver or conversion of surplus carbohydrates to TGs [53]. Many studies have shown that *D. melanogaster* is an appropriate model for mimicking human disease because it exhibits many similarities with mammals, such as conservative metabolic and signal transduction pathways [54]. In *D. melanogaster* larvae, excess dietary sucrose is stored as TGs. When excess fat exceeds the storage capacity of the fat body, it leads to ectopic deposition of free fatty acids into other tissues. This, in turn leads to the development of a metabolic disorder [55,56]. An in vivo experiment revealed that PPYP supplementation significantly reduced the TG content in high-sucrose-fed larvae. Meanwhile, the expression of *SREBP1* and *FAS* was downregulated in the fat body of third instar larvae, and the expression of fatty acid β-oxidation-related genes, including *Acox57D-d*, *FABP*, and *CCT1* was upregulated [57,58]. Our results demonstrated that PPYP supplementation led to a decrease in TG content, which was linked to the reduction in lipogenesis and increase in lipid oxidation in the larval fat body. To summarize, PPYP showed significant lipid-lowering bioactivity under in vitro and in vivo conditions. 

## 4. Materials and Methods 

### 4.1. Materials and Chemicals

*Pyropia yezoensis* was obtained from Seaweed Research Center, National Institute of Fisheries Science (Mokpo, South Korea). The triglycerides (TG) assay kit and total protein (TP) assay kit were provided by Shenzhen Icubio Biomedical Technology Co., Ltd. (Shenzhen, China). TRIzol reagent was purchased from Invitrogen (Carlsbad, CA, USA). TransScript All-in-One First-Strand cDNA Synthesis Supermix for qPCR, and TransStart Top Green qPCR SuperMix were purchased from TransGen Biotech (Beijing, China). Mannose, rhamnose, glucuronic acid, galacturonic acid, glucose, galactose, xylose, arabinose, and fucose were obtained from Sinopharm Chemical Reagent Co. Ltd. (Shanghai, China). Thiazolyl blue tetrazolium bromide (MTT) and dimethyl sulfoxide were purchased from BBI Life Sciences (Shanghai, China). Minimum essential medium (MEM) and fetal bovine serum (FBS) were purchased from Gibco (Grand Island, NE, USA). Penicillin-streptomycin solution (100×) was obtained from Biosharp Life Sciences (Anhui, China). Dextran standards were purchased from China Pharmaceutical Biological Products Analysis Institute (Beijing, China). Palmitic acid (PA) was provided by Kunchuang Biotechnology (Xi’an, China). All other chemical reagents used were of analytical grade. 

### 4.2. Single-Factor Design for Pyropia Yezoensis Polysaccharide

The obtained *P. yezoensis* was air-dried, ground into powder, and passed through a 40-mesh sieve. Subsequently, 500 g powder was refluxed thrice with 2.5 L of 95% ethanol at 60 °C for 2 h to eliminate the alcohol-soluble components. The pretreated powder was then used for the extraction of polysaccharides via the hot water extraction method. Three independent variables, extraction time (from 30 to 240 min), liquid–solid ratio (10:1, 20:1, 30:1, 40:1, 50:1 mL/g), and extraction temperature (from 60 to 100 °C), at three levels were selected for the study, and the yield of polysaccharide extracted (Y) was determined as the response. After the extraction, the filtrate was collected via centrifugation. The filtrate was then precipitated by adding a certain amount of ethanol (95%, *v*/*v*) until the ethanol content dropped to 80% (*v*/*v*). The solution was then stored at 4 °C for 24 h. The solid products (designated as PYP) were then collected via centrifugation, washed thrice with ethanol, and dried at 70 °C overnight. PYP content was determined using the phenol-sulfuric acid method, in which galactose served as the standard [34]. The PYP yield was calculated using Equation (1).
PYP yield (%) = [PYP weight (g)/P. yezoensis pretreated powder(g)] × 100% (1)

### 4.3. Experimental Design 

PYP yield was optimized by a Box–Behnken design (BBD) and using the results of the single-factor experiment with three factors: liquid–solid ratio, time, and temperature. These three factors were designated A, B, and C and categorized into three levels, coded (−1, 0, +1) for low, middle, and high (Table 4). The design enabled us to determine 17 randomly executed experimental points using Design-Expert software (Version 8.0.6). 

### 4.4. Purification of PYP

PYP extraction was performed with optimal extraction parameters (temperature: 100 °C, time: 120 min; liquid–solid ratio: 29.32 mL/g) and extraction process of PYP. The dried PYP was dissolved in distilled water (20 g/L), and the protein was removed using the Sevag method [59]. Then, the aqueous phase was collected, dialyzed, and lyophilized to powder, termed PPYP. The procedure for the extraction and optimization of purified porphyran is illustrated in Figure 7.

### 4.5. Characteristics of PPYP Analysis

The Bradford method was used to determine the protein content [60]. PPYP was analyzed using FT-IR spectroscopy [61]. The sulfate content was determined via the turbidimetric method [62]. Molecular weights and monosaccharide composition were analyzed by high-performance gel permeation chromatography (HPGPC) and high-performance liquid chromatography, respectively, as described previously [28].

### 4.6. Protocol for Obtaining the Palmitic Acid (PA)-Induced HepG2 Cells

#### 4.6.1. Cell Culture and Treatment 

HepG2 cells (a human hepatic carcinoma cell line) were purchased from the Cell Bank of the Shanghai Institute of Biochemistry and Cell Biology. The HepG2 cells were cultured in MEM supplemented with 10% FBS, 100 U/mL penicillin, and 100 µg/mL streptomycin and incubated in a 5% CO2 incubator at 37 °C. The cytotoxicity of PA and PPYP was analyzed via the MTT assay. The HepG2 cells were seeded at a density of 5 × 103 per well with 100 µL of the medium in 96-well plates. PA at a concentration ranging from 25 to 200 µM was used to treat the HepG2 cells for 48 h. PPYP at a concentration ranging from 12.5 to 200 µg/mL was used to treat the HepG2 cells for 48 h. The MTT assay was performed to analyze cell viability. Cell vitality was calculated using Equation (2).
Cell vitality = A treatment/A nontreatment(2)

#### 4.6.2. Triglyceride Content Analysis

The HepG2 cells were seeded in 12-well plates for 48 h; the plates were divided into the control group and those treated with PA (25 to 200 µM) and PA + PPYP (200 µM + 200 µg/mL). These samples were collected using 200 µL of 0.1% Tween-PBS buffer, and centrifuged at 10,000× *g* for 10 min at 4 °C, with three replicates/group. The supernatant was evaluated using an automatic biochemical analyzer purchased from Shenzhen icubio Biomedical Technology Co., Ltd. (Shenzhen, China) equipped with the total protein (TP) assay kit. These samples were heated for 5 min at 70 °C to inactivate endogenous enzymes and then centrifuged to remove debris. Next, the samples were analyzed using the TG assay kit. The assays were repeated thrice for each sample.

#### 4.6.3. Oil Red O Staining

The HepG2 cells were seeded in 24-well plates overnight and then treated with PA (200 µM) or PA + PPYP (200 µM + 200 µg/mL) for 48 h. The HepG2 cells were stained using an oil red O stain kit (for cultured cells) (Solarbio, G1262) according to the manufacturer’s instructions. Images of the stained cells in the normal control group, PA group, and PPYP + PA group were obtained using an inverted microscope (LEICA DMi8, Wetzlar, Germany).

### 4.7. Experimental Protocol for Obtaining High-Sucrose-Fed D. melanogaster Larvae 

#### 4.7.1. *D. melanogaster* Larvae and Treatment

*D. melanogaster w^11^*^8^ was purchased from Qidong Fungene Biotechnology (Jiangsu, China). It was cultured on standard medium (comprising 20 g corn, 8 g sucrose, 8 g yeast, 0.18 g calcium chloride, 1.75 g agar, and 2 mL propionic acid mixed with hot water to make a 250 mL diet) at a constant temperature (25 °C) in a humidity incubator (relative humidity, 70%) with a 12 h light/dark cycle. Next, *D. melanogaster* was kept on standard medium, and embryos were collected on yeasted grape juice agar plates. These embryos were cultured and randomly separated into the normal chow group (NC group), high-sucrose-diet group (HSD group, 1 M sucrose), and high-sucrose-diet group containing 25 mg/mL of PPYP (HSD + PPYP group). 

#### 4.7.2. Triglyceride Content Analysis

The NC, HSD, and HSD + PPYP *D. melanogaster* 3rd instar larvae were collected, ground in 0.1% Tween-PBS buffer (pH 7.4, weight (g): volume (mL) = 1:80), and centrifuged at 10,000× *g* for 10 min at 4 °C, with three replicates/group. The supernatant was evaluated using an automatic biochemical analyzer purchased from Shenzhen Icubio Biomedical Technology Co., Ltd. (Shenzhen, China) equipped with the total protein (TP) assay kit. The samples were heated for 5 min at 70 °C to inactivate endogenous enzymes and then centrifuged to remove debris. Next, the samples were analyzed using a TG assay kit. The assays were repeated three times for each sample.

### 4.8. qPCR Analysis

TRIzol reagent was used to extract total RNA from HepG2 cells or *D. melanogaster* 3rd instar larval fat body (20 larvae/tube). The first-strand cDNA of the HepG2 cells and 3rd instar larvae fat body was synthesized from 1 μg of RNA using the TransScript All-in-One First-Strand cDNA Synthesis Supermix according to the manufacturer’s instructions. The cDNA was analyzed via qPCR using a gene-specific primer (Appendix A). qPCR was performed using LightCycler 96 (Roche, Basel, Switzerland). The initial cycle of the PCR program was completed in 30 s at 95 °C. This was followed by 40 cycles (denaturation at 95 °C for 30 s, annealing at 60 °C for 30 s, and extension at 72 °C for 20 s). The relative quantification of mRNA was performed using the 2^−ΔΔCt^ method. 

### 4.9. Statistical Analysis

Data were calculated as the mean of three replicate determinations with significance at *p* < 0.05, after analyzing the variance (ANOVA) and processing with GraphPad Prism version 8.00 (GraphPad, San Diego, CA, USA). The experimental results of the response surface design were analyzed using the Design-Expert software (version 8.0.6, State-Ease Inc., Minneapolis, MN, USA).

## 5. Conclusions

In this study, the extraction parameters of porphyran from *P. yezoensis* were optimized using the RSM. The optimum extraction conditions were determined as follows: temperature, 100 °C; time, 120 min; liquid–solid ratio, 30 mL/g. PPYP was then obtained using the Sevag method; it was determined to be a β-type sulfated hetero-rhamnogalactan-pyranose. PPYP had a significant inhibitory effect on lipid accumulation in vitro and in vivo, suggesting that PPYP possesses the potential as a dietary supplement for lowering lipid levels in patients with high-sucrose or -fat diet-induced metabolic diseases.

## Figures and Tables

**Figure 1 marinedrugs-19-00053-f001:**
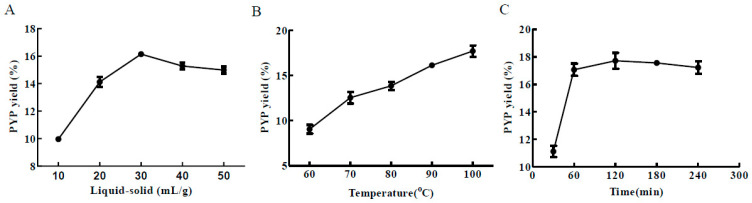
Single-factor experiment analysis of the porphyran (PYP) yield. (**A**) Liquid–solid ratio; (**B**) temperature; (**C**) time.

**Figure 2 marinedrugs-19-00053-f002:**
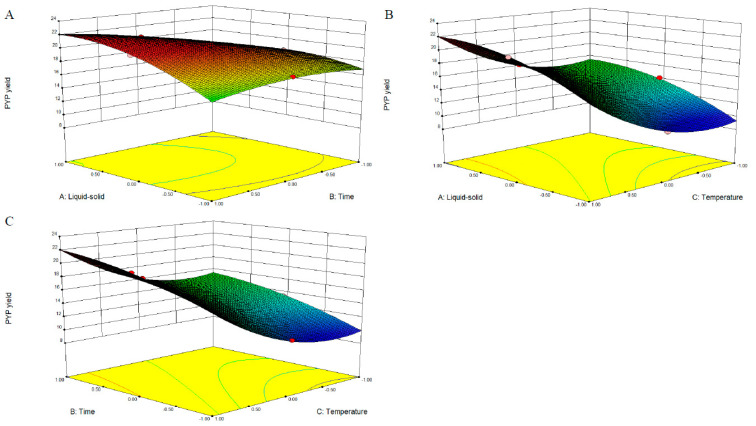
3-D response surface plots of the effects of the various parameters on the PYP yield. (**A**) Liquid–solid ratio and time; (**B**) liquid–solid ratio and temperature; (**C**) time and temperature.

**Figure 3 marinedrugs-19-00053-f003:**
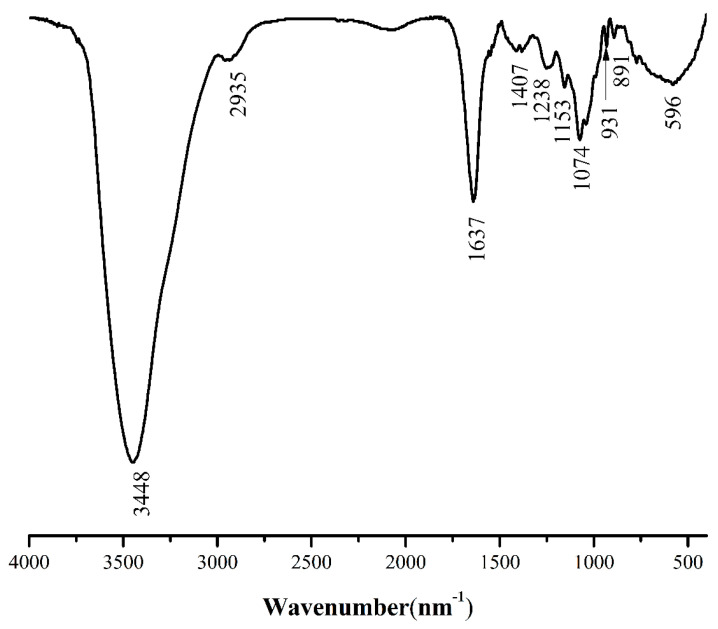
Fourier transform infrared (FT-IR) spectra of PPYP.

**Figure 4 marinedrugs-19-00053-f004:**
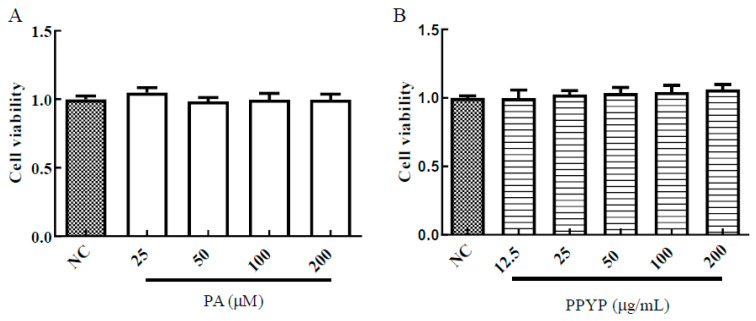
In vitro cell activity of the HepG2 cells incubated with different concentrations of palmitic acid (PA) (**A**) and PPYP (**B**) for 48 h at 37 °C. Data are presented as mean ± SD from three independent experiments.

**Figure 5 marinedrugs-19-00053-f005:**
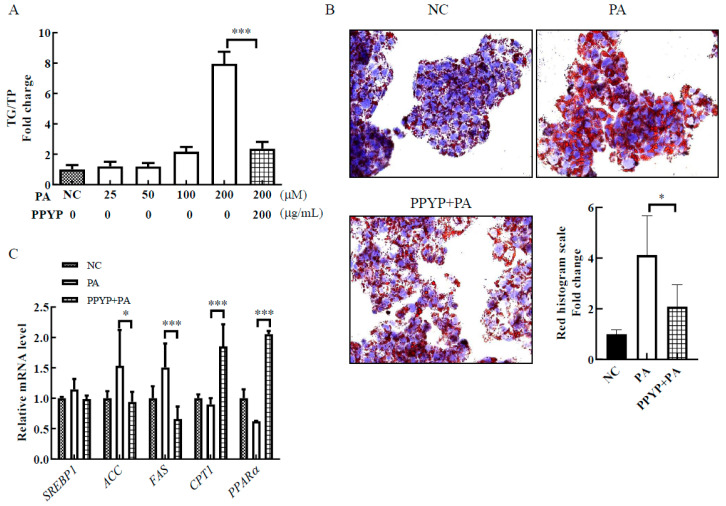
Effect of PPYP on triglyceride (TG) accumulation in PA-induced HepG2 cells. (**A**) TG content; (**B**) Oil red O staining; (**C**) lipid metabolism-related gene expression. Data are presented as mean ± SD from three independent experiments and significance is indicated using stars (*) relative to the comparative group with * *p* < 0.05 and *** *p* < 0.001.

**Figure 6 marinedrugs-19-00053-f006:**
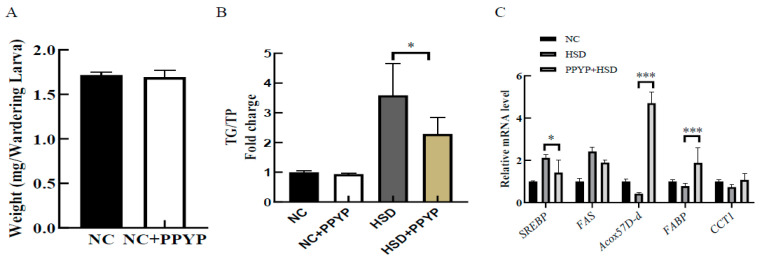
Effects of PPYP on high-sucrose-diet (HSD)-fed 3rd instar larvae. (**A**) Larval weight; (**B**) TG content; (**C**) lipid metabolism-related gene expression. Data are presented as mean ±SD from three independent experiments and significance is indicated using stars (*) relative to the comparative group with * *p* < 0.05 and *** *p* < 0.001.

**Figure 7 marinedrugs-19-00053-f007:**
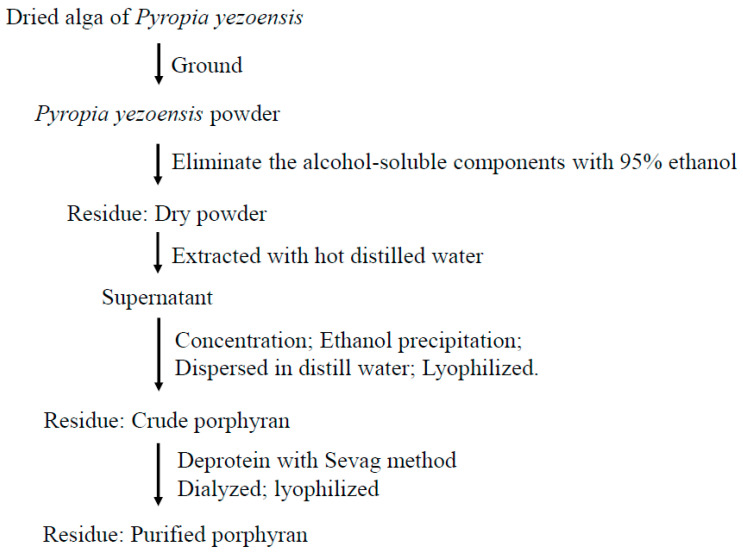
Extraction scheme of *Pyropia yezoensis* porphyran.

**Table 1 marinedrugs-19-00053-t001:** The response values of PYP yield and three variables.

Run	A: Liquid–Solid Ratio (mL/g)	B: Time (min)	C: Temperature (°C)	PYP Yield (%)	
				Actual Value	Predicted Value
1	20 (0)	30 (−1)	100 (1)	18.26	18.40
2	10 (−1)	120 (1)	90 (0)	9.96	10.07
3	20 (0)	75 (0)	90 (0)	14.10	14.34
4	20 (0)	75 (0)	90 (0)	14.27	14.34
5	20 (0)	120 (1)	80 (−1)	14.12	13.98
6	20 (0)	30 (−1)	80 (−1)	11.57	11.48
7	20 (0)	75 (0)	90 (0)	14.48	14.34
8	10 (−1)	30 (−1)	90 (0)	10.67	10.73
9	20 (0)	75 (0)	90 (0)	14.78	14.34
10	30 (1)	75 (0)	100 (1)	20.28	20.25
11	10 (−1)	75 (0)	80 (−1)	10.46	10.49
12	10 (−1)	75 (0)	100 (1)	17.63	17.43
13	30 (1)	30 (−1)	90 (0)	10.74	10.63
14	20 (0)	120 (1)	100 (1)	20.60	20.69
15	20 (0)	75 (0)	90 (0)	14.05	14.34
16	30 (1)	120 (1)	90 (0)	16.15	16.09
17	30 (1)	75 (0)	80 (−1)	13.37	13.57

**Table 2 marinedrugs-19-00053-t002:** ANOVA of the response surface model for PYP yield.

Source	Sum of Squares	Df	Mean Square	F-Value	*p*-Value
Model	172.93	9	19.21	254.25	***
A-Liquid–solid	17.46	1	17.46	231.10	***
B-Time	11.50	1	11.50	152.12	***
C-Temperature	92.82	1	92.82	1228.26	***
AB	9.36	1	9.36	123.91	***
AC	0.017	1	0.017	0.22	0.6507
BC	0.011	1	0.011	0.15	0.7138
A^2^	10.50	1	10.50	138.96	***
B^2^	3.24	1	3.24	42.83	***
C^2^	30.20	1	30.20	399.65	***
Residual	0.53	7	0.076		
Lack of fit	0.17	3	0.056	0.63	0.6344
Pure error	0.36	4	0.090		
Correlation total	173.46	16			
R^2^	0.9970		R^2^adj	0.9930	
C.V.%	1.90		Pred R-Squared	0.9811	

*** *p* < 0.001.

**Table 3 marinedrugs-19-00053-t003:** Physicochemical properties of PPYP.

Name	Sugar (%)	Sulfate (mmol/mL)	3,6-anhydro-α-ʟ-galactose (%)	Molecular Weight (kDa)	Monosaccharide Molar Ratio Gal Rha	
PPYP	93.2 ± 1.5	1.2 ± 0.03	20.8 ± 1.1	151.6	5.3	1.0

Note: Gal: galactose; Rha: rhamnose. Data are presented as the means ± SD.

**Table 4 marinedrugs-19-00053-t004:** Level and coded values of different variables in the Box–Behnken design (BBD).

Symbols	Independent Variable	Level		
		−1	0	1
A	Liquid–solid ratio (mL/g)	10:1	20:1	30:1
B	Time (min)	30	75	120
C	Temperature (°C)	80	90	100

## Data Availability

Data is contained within the article or Appendix A.

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
