# Peer review of "Optimization of Porphyran Extraction from Pyropia yezoensis by Response Surface Methodology and Its Lipid-Lowering Effects"

_marinedrugs, 2021, doi:10.3390/md19020053_

Round 1

Reviewer 1 Report

To the authors of the manuscript.

I have found of interest that PPYP inhibits lipid accumulation in the in vitro models that you have tested. However, I have missed some experiments to further characterize this response so that I cannot recomment its publication at the current status.

Please find attached a list with some concerns that have led me to take this decission:

  • English should be further revised as sometimes it is difficult to follow. Most part of the sentences are misleading and need to be reformulated.
  • NAFLD should be better introduced. You have not mentioned the second hits that occur during the progression of the diseases and you have not characterized the effect of PYPP in NAFLD, just in steatosis. A more complete characterization of its effect should include additional experiments such as the determination of ROS production, the development of ER stress, mitochondrial dysfunction markers... 
  • In vivo experiments would be desirable before proposing PYPP as a supplement for NAFLD.
  • The presentation of results is really confusing. E.g. the Figure 1 does not appear and you use a complete table (Table 4) to include only one file.

Author Response

Response to the reviewer’s comments
1. English should be further revised as sometimes it is difficult to follow. Most part of the sentences are misleading and need to be reformulated.
Response:
Thank you so much for your comments and suggestions those are highly appreciated.
According to the suggestion of the reviewer, we revised the language and grammar thoroughly.
2. NAFLD should be better introduced. You have not mentioned the second hits that occur during the progression of the diseases and you have not characterized the effect of PPYP in NAFLD, just in steatosis. A more complete characterization of its effect should include additional experiments such as the determination of ROS production, the development of ER stress, mitochondrial dysfunction marker…
Response:
Nonalcoholic fatty liver disease (NAFLD) is characterized by two steps of liver injury: intrahepatic lipid accumulation (hepatic steatosis), and inflammatory progression to nonalcoholic steatohepatitis (NASH) (the 'two-hit' theory). [1-3] Here, we evaluated the effect of PPYP on the inhibition of lipid accumulation in the early stage of NAFLD. Our present results demonstrated that PPYP showed a significant lipid-lowering effect in vitro and in vivo. Therefore, based on the present data, we have removed the inappropriate presentation that PPYP could treat NAFLD in the revised manuscript. Finally, the potential application of PPYP in improving NAFLD was supplemented in the Discussion section.
Reference
[1] Perlemuter, G.; Bigorgne, A.; Cassard-Doulcier, A.M.; Naveau, S. Nonalcoholic fatty liver disease: from pathogenesis to patient care. Nat. Clin. Pract. Endocrinol. Metab. 2007, 3, 458–469.
[2] Tessari, P.; Coracina, A.; Cosma, A.; Tiengo, A. Hepatic lipid metabolism and non-alcoholic fatty liver disease. Nutr. Metab. Cardiovasc. Dis. 2009, 19, 291–302.
[3] Lim, J. S.; Mietus-Snyder, M.; Valente, A.; Schwarz, J.M.; Lustig, R.H. The role of fructose in the pathogenesis of NAFLD and the metabolic syndrome. Nat. Rev. Gastroenterol. Hepatol. 2010, 7, 251–264.
3. In vivo experiments would be desirable before proposing PPYP as a supplement for NAFLD.
Response:
Accumulated evidence has shown that Drosophila melanogaster is a reliable model organism for the research on metabolic disease, as it exhibited many similarities with mammalian such as conservative metabolic and signal transduction pathways. [1, 2] Our data showed that PPYP could significantly inhibit the lipid accumulation in high-sucrose-fed Drosophila melanogaster. In addition, we are also preparing to further confirm the lipid-lowering effect of PPYP in a mice model.
Reference
[1] Koyama, T.; Texada, M.J.; Halberg, K.A.; Rewitz, K. Metabolism and growth adaptation to environmental conditions in Drosophila. Cell. Mol. Life Sci.: CMLS, 2020, 77, 4523–4551.
[2] Ugur, B.; Chen, K.; Bellen, H.J. Drosophila tools and assays for the study of human diseases. Dis. Model Mech. 2016, 9, 235–244.
4. The presentation of results is really confusing. E.g. the Figure 1 does not appear and you use a complete table (Table 4) to include only one file.
Response:
It has been revised accordingly.

Reviewer 2 Report

This manuscript describes extraction methods for phytochemicals from Pyropia yezoensis porphyran (PYP) including testing of extraction perameters (time, temp, etc.).  It also investigates the effects of these on triglyceride production in HepG2 cells and D. melanogaster.

minor writing recommendations:

There are several awkward sentences including (but not limited to :

line 16: "In this study, using response surface methodology was employed to 16 investigate the effects of liquid-solid ratio, extraction temperature, and extraction time on Pyropia 17 yezoensis porphyran (PYP) yield."

line 289: Therefore, PPYP has the potential to be 9 used a drug to treat NAFLD.

Other recommendations:

line 88: needs a space at 120 min. "120min"

line 116/117: be consistent with writting 3-D or 3D.  "the 3D response 117 surface and 2-D contour plots"

line 124:  remove an extra "." at "C."

Figure 5 legend refers to statistics but there are non given on the graph (no ns or anything else). "Each group contained three independent groups and significance is 179 indicated using stars (*) relative to the comparative group with *p<0.05 and ***p<0.001; ns stands 180 for no significant difference."

Figure 6. Legend is incorrect.  Are panels A and B switched?? 

Author Response

Response to the reviewer’s comments
1. There are several awkward sentences including (but not limited to:
Line 16: “In this study, using response surface methodology was employed to 16 investigate the effects of liquid-solid ratio, extraction temperature, and extraction time on Pyropia 17 yezoensis porphyran (PYP) yield.”
Line 289: Therefore, PPYP has the potential to be 9 used a drug to treat NAFLD.
Response:
According to the suggestion of the reviewer, we revised the language and grammar thoroughly. We rephrase the colored sentences, but, of course, keep the meaning of these.
Other recommendations:
2. Line 88: needs to a space at 120 min. “120 min”
Response:
In line 88, it has been revised accordingly.
3. Line116/117: be consistent with writing 3-D or 3D. “the 3D response 117 surface and 2-D contour plots”
Response:
In line 116/117, the “3D” should be changed into “3-D”. It has been confirmed accordingly.
4. Line 124: remove an extra “.” at “C.”
Response:
In line 124, extra “.” at “C.” has been deleted accordingly.
5. Figure 5 legend refers to statistics but there are non given on the graph (no ns or anything else). “Each group contained three independent groups and significance is 179 indicated using stars (*) relative to the compared group with *p<0.05 and ***p<0.001; ns stands 180 for no significant difference.”
Response:
It has been revised accordingly. And Figure 5 legend has been clarified describe according to the manuscript.
6. Figure 6. Legend is incorrect. Are panels A and B switched??
Response:
It has been revised in Figure 6 legend accordingly.

Reviewer 3 Report

In this manuscript, the authors describe the effects of liquid-solid ratio, extraction temperature, and extraction time on Pyropia yezoensis porphyran yield using response surface methodology. This study demonstrated that the finding of Pyropia yezoensis porphyran could be applied in the development of effective therapeutics to protect against lipid accumulation in the liver. The findings are important to optimize drug development of liver protection applying surface response methodology.

Overall, the MS is written well; however, a minor revision is required following the comments below.

  • Page 1, lines 32-36: The authors mentioned “Marine algae are abundant sources of medicinal phytonutrients with various benefits for human health. Among them, sulfated polysaccharides have excellent biological features including antioxidant, anti-inflammatory, antitumor, and immunomodulatory activities [ref 1]. These polysaccharides were found in different macroalgae, including brown algae, red algae, and green algae [ref 2]”. The readers might be curious to know about other algae such as a high abundance species of Coralline algae (e.g., https://doi.org/10.1038/srep06162). A short comparative discussion about marine algae could help the readers to understand the present study.
  • FT-IR spectroscopy characteristics of PPYP: The authors didn’t explain the band 1637. It is a strong band so it is required to include in the result section.

Author Response

Response to the reviewer’s comments

1. Page 1, lines 32-36: The authors mentioned “Marine algae are abundant sources of medicinal phytonutrients with various benefits for human health. Among them, sulfated polysaccharides have excellent biological features including antioxidant, anti-inflammatory, antitumor, and immunomodulatory activities [ref 1]. These polysaccharides were found in different macroalgae, including brown algae, red algae, and green algae [ref 2]”. The readers might be curious to know about other algae such as a high abundance species of Coralline algae (e.g., https://doi.org/10.1038/srep06162). A short comparative discussion about marine algae could help the readers to understand the present study.
Response:
A short comparative discussion for the various algae was supplemented in the Introduction section according to the suggestion of the reviewer.
2. FT-IR spectroscopy characteristics of PPYP: The authors didn’t explain the band 1637. It is a strong band so it is required to include in the result section.
Response:
The brief description for the band 1637 was supplemented in 2. Results (2.4. FT-IR spectroscopy characteristics of PPYP) according to the suggestion of the reviewer.

Round 2

Reviewer 1 Report

To the authors of the manuscript,

After having revised this version of the manuscript, I have found that you have followed the suggestions improving the quality of the paper. However, I still consider that some changes should be performed before its publication. Therefore, I have recommended MAJOR REVISIONS prior its acceptance.

Please find a list with the concerns:

  • I have guessed that PA is palmitic acid because it is indicated nowhere. You should include each abbreviature carefully to avoid this to the reader.
  • I miss some qPCR in vitro and anothers in vivo because you have analyzed different genes. I would thank an unification of the genes chosen.
  • I such PCR you mention that FAS decreases significantly and you do not indicate in Fig. 6C. Please check carefully the relationship between the graphs and the text as they are misleading.
  • The relevance of the genes involved in lipid metabolism should be analyzed. I suggest to select specific inhibitors for the regulated genes and perform a proof of concept to observe if PPYP exerts the same effect.
  • I still miss mice experiments as I am "skeptical" about considering D. melanogaster as a complete in vivo model.

Author Response

Response to the reviewer’s comments

  1. I have guessed that PA is palmitic acid because it is indicated nowhere. You should include each abbreviature carefully to avoid this to the reader.

Response:

Thank you very much for all the comments that help us a lot to improve the quality of this manuscript.

Each abbreviature has been double-checked and revised accordingly.

  1. I miss some qPCR in vitroand anothers in vivo because you have analyzed different genes. I would thank an unification of the genes chosen.

Response:

We think that PPYP shows a lipid-lowering effect by regulating the expression of lipid metabolism-related genes reducing lipogenesis and increasing fatty acid β-oxidation. Many studies have reported that carnitine palmitoyltransferase 1 (CPT-1) and peroxisome proliferator-activated receptor alpha (PPARα) are important components of the fatty acid β-oxidation pathway in mammals. Their expression levels are commonly used to evaluate fatty acid β-oxidation in mammals [1-4]. In Drosophila melanogaster, both 57D distal acyl coenzyme A oxidase (Acox57D-d) and FABP (fatty acid binding protein) are considered to be major components of fatty acid β-oxidation [5,6]. It has been reported that their expression levels were used to evaluate fatty acid β-oxidation in Drosophila [7,8]. Thus, we examined the expression levels of these genes according to the reported studies.

Reference

[1] Mun, J.; Park, J.; Yoon, H.G.; You, Y.; Choi, K.C.; Lee, Y.H.; Kim, K.; Lee, J.; Kim, O.K.; Jun, W. Effects of Eriobotrya japonica water extract on alcoholic and nonalcoholic fatty liver impairment. J. Med. Food, 2019, 22, 1262–1270.

[2] Zhang, Q.; Kong, X.; Yuan, H.; Guan, H.; Li, Y.; Niu, Y. Mangiferin improved palmitate-induced-insulin resistance by promoting free fatty acid metabolism in HepG2 and C2C12 cells via PPARα: Mangiferin improved insulin resistance. J. Diabetes Res. 2019, 2019, 2052675.

[3] Schreurs, M.; Kuipers, F.; van der Leij, F.R. Regulatory enzymes of mitochondrial beta-oxidation as targets for treatment of the metabolic syndrome. Obes. Rev. 2010, 11, 380–388.

[4] Park, M.Y.; Mun, S.T. Dietary carnosic acid suppresses hepatic steatosis formation via regulation of hepatic fatty acid metabolism in high-fat diet-fed mice. Nutr. Res. Pract. 2013, 7, 294–301.

[5] Faust, J.E.; Verma, A.; Peng, C.; McNew, J.A. An inventory of peroxisomal proteins and pathways in Drosophila melanogasterTraffic 2012, 13, 1378–1392.

[6] Lee, S.H.; Lee, S.K.; Paik, D.; Min, K.J. Overexpression of fatty-acid-β-oxidation-related genes extends the lifespan of Drosophila melanogasterOxid. Med. Cell. Longev. 2012, 2012, 854502.

[7] Kayashima, Y.; Murata, S.; Sato, M.; Matsuura, K.; Asanuma, T.; Chimoto, J.; Ishii, T.; Mochizuki, K.; Kumazawa, S.; Nakayama, T.; Yamakawa-Kobayashi, K. Tea polyphenols ameliorate fat storage induced by high-fat diet in Drosophila melanogasterBiochem. Biophys. Rep. 2015, 4, 417–424.

[8] Palanker, L.; Tennessen, J.M.; Lam, G., Thummel, C.S. Drosophila HNF4 regulates lipid mobilization and beta-oxidation. Cell Metab. 2009, 9, 228–239.

  1. I such PCR you mention that FAS decreases significantly and you do not indicate in Fig. 6C. Please check carefully the relationship between the graphs and the text as they are misleading.

Response:

It has been revised accordingly.

  1. The relevance of the genes involved in lipid metabolism should be analyzed. I suggest to select specific inhibitors for the regulated genes and perform a proof of concept to observe if PPYP exerts the same effect.

Response:

Our present data can only prove that PPYP affected lipid metabolism-related genes and exerted lipid-lowering effect. There is no further experimental evidence to reveal the direct targets of PPYP. Our research plan is to screen the potential targets of PPYP and revealing molecular mechanism.

  1. I still miss mice experiments as I am "skeptical" about considering melanogaster as a complete in vivomodel.

Response:

Many metabolic pathways are highly conserved in both human and D. melanogaster. Thus, D. melanogaster is widely used to replicate human diseases, such as obese, diabetes and cardiovascular disease, for revealing molecular mechanisms, high-throughput drug screens as well as new target discovery [1-4]. Thus, we here use D. melanogaster to replicate diet-induced disorders of lipid metabolism to evaluate the lipid-lowering effect of Pyropia yezoensis porphyran.

Reference

[1] Pandey, U.B.; Nichols, C.D. Human disease models in Drosophila melanogaster and the role of the fly in therapeutic drug discovery. Pharmacol. Rev. 2011, 63, 411–436.

[2] Schneider D. Using Drosophila as a model insect. Nat. Rev. Genet. 2000, 1, 218–226.

[3] Cheng, L.; Baonza, A.; Grifoni, D. Drosophila Models of Human Disease. BioMed Res. Int. 2018, 2018, 7214974.

[4] Lüersen, K.; Röder, T.; Rimbach, G. Drosophila melanogaster in nutrition research-the importance of standardizing experimental diets. Genes Nutr. 2019, 14, 3.

Round 3

Reviewer 1 Report

To the authors of the manuscript,

I sincerelly do not appreciate that you have performed any modifications of which I suggested. I mainly base on the fact that I suggest to perform at least the same qPCR in HepG2 to Drossophila and it is not realized. Thus, I still have recommended MAJOR REVISIONS.

I strongly recommend to take some days (or weeks) to better answer this report before sending another version.

Best regards.

This manuscript is a resubmission of an earlier submission. The following is a list of the peer review reports and author responses from that submission.